# Research on Scale Improvement of Geochemical Exploration Based on Remote Sensing Image Fusion

Haifeng Ding [1,2], Linhai Jing [1,2], Mingjie Xi [3,*], Shi Bai [4], Chunyan Yao [5] and Lu Li [6]

1 International Research Center of Big Data for Sustainable Development Goals, Beijing 100094, China
2 Key Laboratory of Digital Earth, Aerospace Information Research Institute, Chinese Academy of Sciences, Beijing 100094, China
3 Institute of Geophysical and Geochemical Exploration, Chinese Academy of Geological Sciences, Langfang 065000, China
4 Nanhu Laboratory, Jiaxing 314001, China
5 Nanjing Center of China Geological Survey, Nanjing 210016, China
6 National Geological Library of China, Geoscience Documentation Center of China Geological Survey, Beijing 100083, China
* Correspondence: xmingjie@mail.cgs.gov.cn; Tel.: +86-10-82178079

**Abstract:** Both remote sensing and geochemical exploration technologies are effective tools for detecting target objects. Although information on anomalous geochemical elemental abundances differs in terms of professional attributes from remote sensing data, both are based on geological bodies or phenomena on the Earth's surface. Therefore, exploring the use of remote sensing data with high spatial resolution to improve the accuracy of small-scale geochemical data, and fusing them to obtain large-scale geochemical layers could provide new data for geological and mineral exploration through inversion. This study provides a method of fusing remote sensing images with small-scale geochemical data based on a linear regression model that improves the resolution of geochemical elemental layers and provides reference data for mineral exploration in areas lacking large-scale geochemical data. In the Xianshuigou area of Northwest China, a fusion study was conducted using 200,000 geochemical and remote sensing data. The method provides fused large-scale regional chemical data in well-exposed areas where large-scale geochemical data are lacking and could provide potential data sources for regional mineral exploration.

**Keywords:** geochemical exploration; remote sensing; image fusion; mineral exploration

## 1. Introduction

Geochemical data play one of the most important roles in mineral exploration and environmental pollution studies [1–4]. Geochemical data play a significant role in mineral exploration. They provide information about the chemical composition of rocks, soils, and sediments which can be used to identify mineral deposits that may not be visible on the Earth's surface. Geochemical surveys can help identify areas with anomalous concentrations of metals and minerals, which can be indicative of mineral deposits. These surveys involve collecting and analyzing samples of rocks, soils, and water to determine the presence and concentration of certain elements. Its application in mineral exploration can lead to the discovery of economically viable mineral deposits and can contribute to the growth of the mining industry. The United Nations Educational, Scientific, and Cultural Organization (UNESCO) International Centre on Global-Scale Geochemistry was established in 2016 in China. The global geochemistry network, benchmark chemical elements, and China's geochemistry observation network have been gradually established with the support of this center. Due to the limitations of terrain and the cost of manual acquisition, the sampling points for geochemical elemental abundances are usually insufficient. According to the current situation of geochemical exploration in China, the scale is generally

1:500,000, 1:250,000, or 1:200,000. The sampling distance resolution is at least 1 km × 1 km. The low sampling point density leads to the low resolution of the geochemical elemental abundances, and due to the massive workload of 1:50,000 geochemical explorations, this density is far from sufficient to cover most of China. In many works, only small-scale and low-resolution geochemical data can be used for reference. With existing technology, if the resolution of geochemical elemental layers is to be improved, the sampling density of geochemical elemental abundance sampling points must increase, which will not only greatly increase the cost of manual sampling but also increase the difficulty of sampling due to the complex terrain in some areas.

Based on the spectral reflectance features from the visible (VIS) to visible and near-infrared (VNIR), shortwave infrared (SWIR), and thermal infrared (TIR) wavelengths, many alteration minerals, such as carbonates, sulfates, hydroxides, oxides, quartz, olivine, feldspar, and silicate minerals, have been well identified and mapped [5]. Synthetic aperture radar (SAR) can be used in vegetation cover areas because its microwave radiation penetrates clouds and vegetation [6]. For these features, the VIS, VNIR, SWIR, and TIR images are generally more useful for lithological mapping, alteration information, and mineral deposits. However, radar images are more useful for structural mapping, surface and subsurface geomorphological features, and roughness levels of different rocks.

Both remote sensing and geochemical exploration technologies are very effective methods for detecting target objects [7,8]. Although information on anomalous geochemical elemental abundances and that which is provided by remote sensing data are different in terms of their professional attributes, they are both obtained based on geological bodies or geological phenomena on the Earth's surface, and they are both geological information that include different forms of physical and chemical properties related to mineralization; therefore, these two sources of information must be correlated, which is confirmed by many studies. High anomalies in geochemical elemental abundance data indicate the enrichment of geochemical elements, while remote sensing alteration information indicates the type and intensity of hydrothermal alteration at the surface, which is directly related to the enrichment of elements [9–11].

To solve the problem of large-scale geochemical data using remote sensing images, this paper provides an image fusion method based on a regression model to improve the resolution of geochemical elemental layers by fusing small-scale geochemical data and remote sensing images. The large-scale geochemical data that were obtained from this fusion process may not be comparable to the real geochemical mapping data, but they can at least provide finer and richer information on the details of geochemical anomalies. Thus, the purpose of improving the resolution of geochemical elemental layers without increasing the cost and difficulty of manual sampling is achieved.

## 2. Geological Setting

The Xianshuigou area (Figure 1), which is located in the Beishan orogenic belt, is part of the Ejina Banner, Inner Mongolia Autonomous Region, China, and is at the intersection of the Kazakhstan, Tarim, and North China plates along the southern margin of the Central Asian Orogenic Belt (CAOB). The orogenic belt is composed of several arc belts that are separated by ophiolite belts, arcs, and blocks formed by subduction–accretion between the Tarim craton and the Kazakhstan plate [12–18]. The Beishan orogenic belt has several complete ophiolites that probably formed during the early Paleozoic in southern China. There are three major ophiolitic belts that are distributed from north to south: the Hongshishan-Baiheshan, Shibanjing-Xiaohuangshan, Yueyashan-Xichangjing, and Liuyuan-Zhangfangshan ophiolitic belts [17,19,20]. According to the spatial and temporal distributions and rock assemblage characteristics of the ophiolite belts, the Xianshuigou area can be divided into the Huaniushan magmatic arc, Gongpoquan-Qiyishan magmatic arc, Heiyingshan magmatic arc, Hanshan microcontinent, and Queershan magmatic arc from south to north [20].

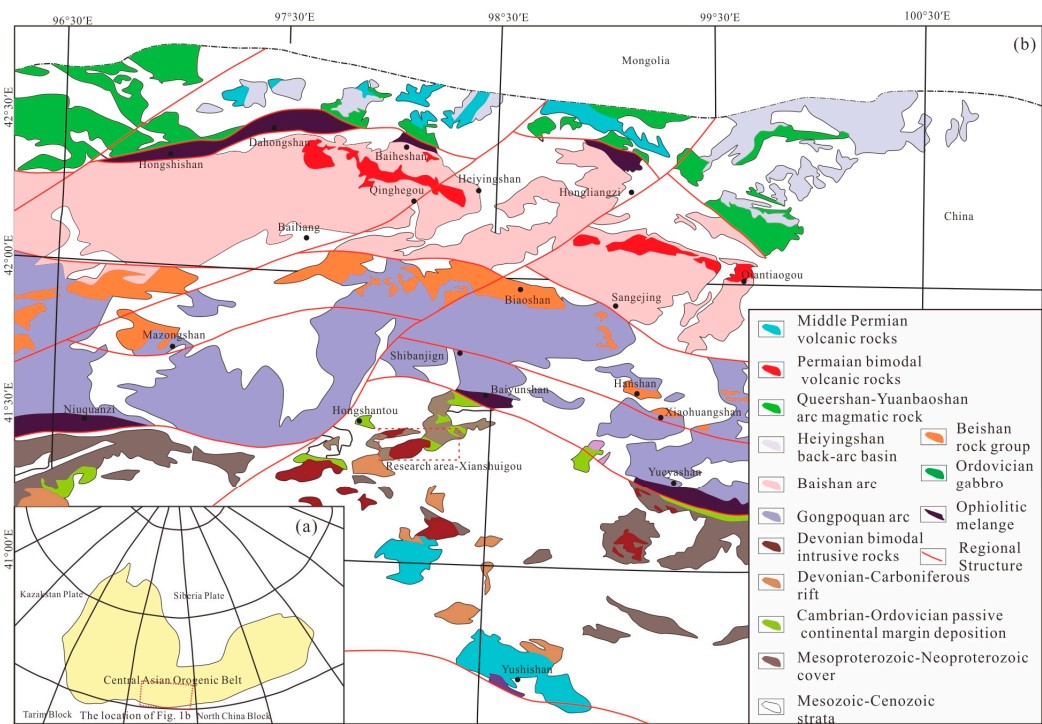

**Figure 1.** The regional geological map (**a**,**b**).

Paleozoic intrusive rocks are widely developed in the Xianshuigou area and mainly include early Silurian quartz amphibolite, tonalite, granite amphibolite, and diorite formed in an island arc environment; Early Devonian medium-grained porphyritic orthogranite, diorite granite, and quartz diorite formed in a post-collisional orogenic environment; and Late Devonian gabbro, quartz amphibolite, tonalite, and granite amphibolite. Stratigraphically, the Paleozoic Beishan Group is mainly exposed in the northern part of the Shibanjing-Xiaohuangshan tectonic zone; the Ordovician–Silurian Gongpoquan Group, with an island arc volcanic nature, is mainly distributed in the southern part of the tectonic zone; and the Early–Middle Triassic Erduanjing Group and the Early Cretaceous Chijinbao Group are mainly distributed in the northwestern part of the study area and are in angular unconformity contact with the underlying geological bodies.

The study area is also located in the Beishan metallogenic belt, which is an important part of the CAOB metallogenic belt and has an excellent geological background for mineralization. Several mineralization zones are developed in the Beishan metallogenic belt from north to south, namely the Hongshishan-Baiheshan-Pengboshan, Jijitaizi, Hongliuhe-Baiyunshan-Yueyashan, and Huitongshan mineralization zones. In particular, the Beishan metallogenic belt is a favorable area for finding copper polymetallic deposits. A series of copper polymetallic deposits, such as the Sandaominshui, Gaoshishan, Elegenwula, and Liushashan deposits, have been found in the Beishan area of Inner Mongolia. The geochemical investigation results show that the geochemical anomalies of copper-based metallogenic elements in the area are characterized by large scales, high intensities, and obvious zoning. Therefore, there are not only copper polymetallic deposits in the area but also a large amount of valuable information for mineral searches. However, the number of copper polymetallic deposits found in the region is small compared to those in the North Mountain mineralized belt in Xinjiang and Gansu Provinces.

## 3. Data

### 3.1. Geochemical Data

The Regional Geochemistry–National Reconnaissance (RGNR) has been carried out since 1978 and covers more than seven million square kilometers of China's territory at scales of 1:500,000, 1:250,000, or 1:200,000. In the program, the work method specification,

sampling methods of stream sediments and rocks, multimethod analytical schemes, and certification of standard reference materials were developed at additional scales (1:100,000, 1:50,000, etc.) for global standard works (http://www.cgs.gov.cn).

In this study, 1:200,000 geochemical data were used as low-scale layers, which compose the database of the China Geological Survey. Geochemical data at a scale of 1:50,000 were used as high-scale layers to verify the fusion results which were collected by the Institute of Geophysical and Geochemical Exploration, Chinese Academy of Geological Sciences, in the Heiyingshan Geological and Mineral Survey Program supported by the China Geological Survey.

All the samples were sieved, and the <80 μm fraction was subjected to inductively coupled plasma–optical emission spectrometry (ICP–OES) for the determination of 44 elements, except Au.

### 3.2. Remote Sensing Data

This section may be divided by subheadings. It should provide a concise and precise description of the experimental results, their interpretation, as well as the experimental conclusions that can be drawn.

In this study, the Advanced Spaceborne Thermal Emission and Reflectance Radiometer (ASTER) and Sentinel-2 remote sensing images were used for fusion. ASTER data were downloaded from the United States Geological Survey (USGS) website (https://glovis.usgs.gov/), and Sentinel-2 data were downloaded from the open hub of the European Space Agency (ESA) website (https://scihub.copernicus.eu).

The ASTER on the Terra platform, launched on December 18 1999, provides enhanced mineral mapping capabilities for remote sensing geology because of its additional SWIR band settings [21–25]. The ASTER data consist of three separate subsystems with a total of 14 spectral bands, including three VIS-VNIR bands (0.52–0.86 μm) with a 15 m spatial resolution, six SWIR bands (1.60–2.43 μm) with a 30 m spatial resolution, and five TIR bands (8.12–11.65 μm) with a 90 m spatial resolution [26,27]. There are several levels of ASTER data, and the ASTER Level 1 Precision Terrain Corrected Registered At-Sensor Radiance (AST_L1T) data can be freely downloaded from the USGS Global Visualization Viewer (GloVis), which also contains calibrated at-sensor radiance (https://lpdaac.usgs.gov/). The Fast Line-of-Sight Atmospheric Analysis of Spectral Hypercube (FLAASH) algorithm was used in this study for the atmospheric correction of ASTER data using ENVI 5.3 software.

To address issues related to environmental monitoring, the European Commission (EC) and European Space Agency (ESA) have established the Global Monitoring for Environment and Security (GMES) and the European Earth Observation program Copernicus [28]. The space component of the program consists of satellite missions labeled Sentinel-1 to 5 [29,30]. The Sentinel-2 Multispectral Instrument (MSI) has 13 spectral bands from the VNIR (ten bands) to the SWIR bands (three bands), four bands with 10 m spatial resolution, six bands with 20 m spatial resolution, and three bands with 60 m spatial resolution. Sentinel-2 provides high-resolution and wide-range multispectral data with a revisit time of 5 days [31].

## 4. Methods

### 4.1. Remote Sensing

The ASTER scene includes Level-1T format data that have already been used for radiometric calibration, geometric calibration, and orthorectified calibration. The 30 m SWIR bands were resampled to a 15 m pixel size to generate 9-band stacked datasets of the VNIR and SWIR bands. In addition, crosstalk correction was applied to bands 4, 5, and 9 for energy spillover errors [32]. For atmospheric correction, the model fast line-of-sight atmospheric analysis of spectral hypercubes (FLAASH) module was employed to eliminate atmospheric transmittance, solar irradiance topographic effects, and albedo effects by ENVI version 5.3 [33–36]. ASTER data have a rich design of shortwave infrared bands, which can be used for a variety of remotely sensed alteration information extraction. Figure 1 shows

the ASTER band 741 pseudo-color composite map (Figure 2), from which it can be clearly seen that different geological units are displayed as different colors.

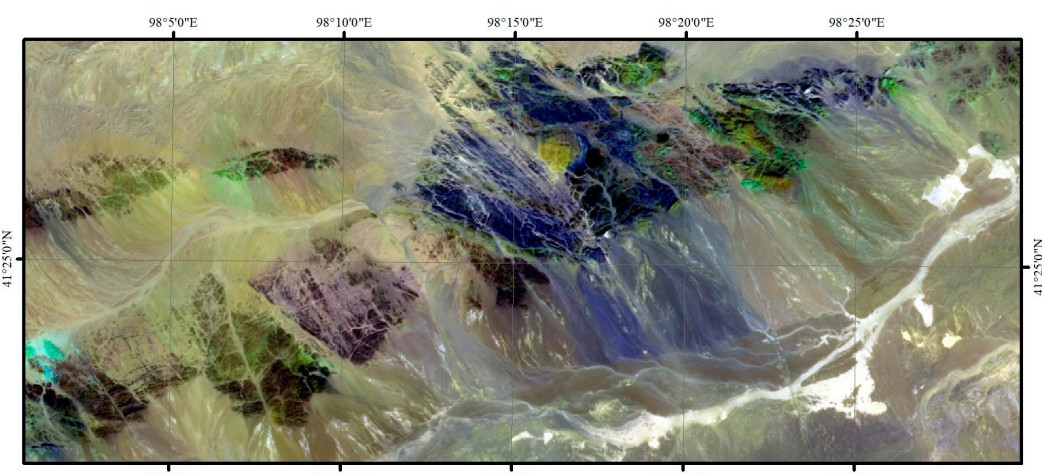

**Figure 2.** The ASTER image (ASTER red, green, and blue: bands 7, 4, and 1, respectively).

The Level-1C data of Sentinel-2 were freely downloaded from the ESA website, which provides orthorectified top-of-atmosphere (TOA) reflectance with subpixel multispectral registration. The Sen2Cor is a processor for Sentinel-2 Level 2A product generation and formatting, which creates bottom-of-atmosphere, optional terrain- and cirrus-corrected reflectance images, as well as aerosol optical thickness, water vapor, scene classification maps, and quality indicators for cloud and snow probabilities [37]. With a spatial resolution of 10 m, Sentinel-2 has the highest spatial resolution of any data that can be acquired free and unrestricted. Using Sentinel-2 band 842 pseudo-color composite maps (Figure 3), it is possible to see finer delineations and distinctions in the interior of geological bodies that cannot be shown in ASTER.

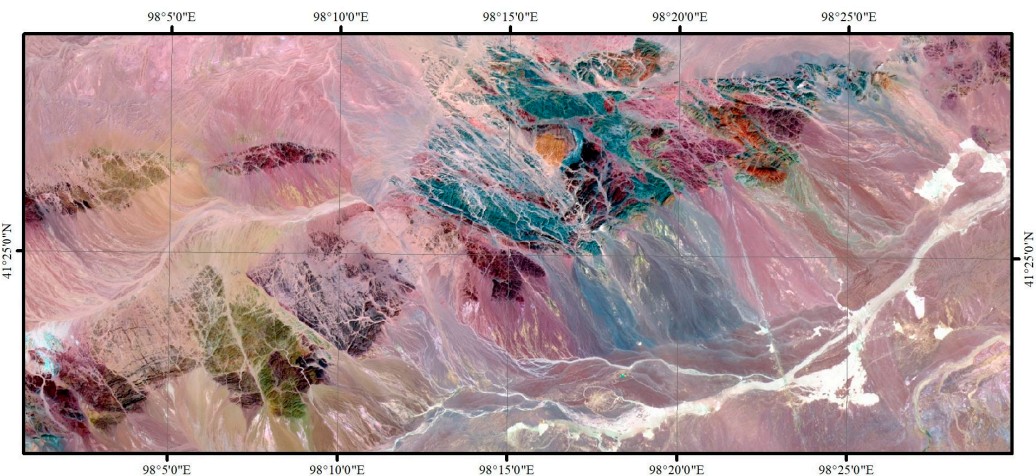

**Figure 3.** The Sentinel-2 image (Sentinel-2 MSI red, green, and blue: bands 8, 4, and 2, respectively).

### 4.2. Geochemical Data

The fusion geochemical data used to extract geochemical anomalies in this study are geochemical survey data at a scale of 1:200,000. A scale of 1:50,000 for geochemical data was used to verify the accuracy of the fusion results. The samples were 0–20 cm weathered surfaces that overlie the stratigraphic units and exposed range of intrusive rocks. These samples were collected within 100 m/25 m of the sampling point, with one sample consisting of three to five points, which were then naturally air-dried and screened indoors for measurement. The distribution of sampling points is shown in Figures 4 and 5.

According to the major deposit types, we selected Cu for data processing and analysis. The spatial distribution of Cu with the Kriging interpolation method is shown in Figure 6 (200,000) and Figure 7 (50,000). The 50,000 geochemical sampling was divided into debris and soil. For better integration studies, only the debris sampling area on the west side of the study area was considered when referring to the 50,000 geochemical data.

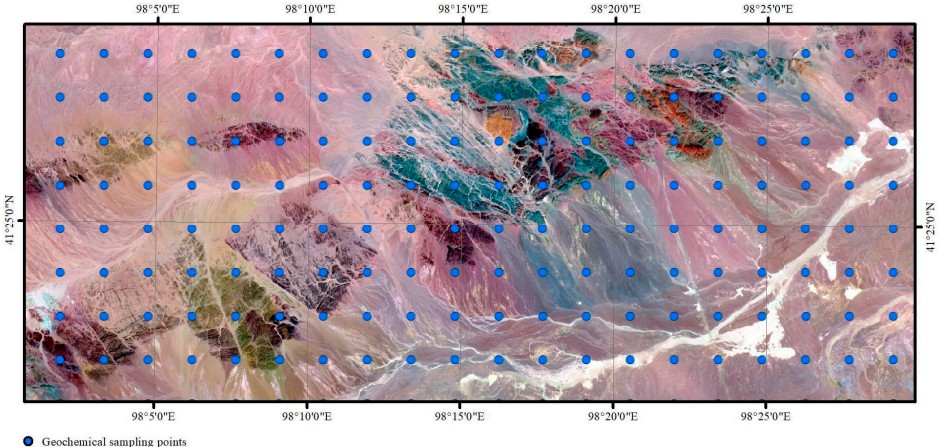

**Figure 4.** The 200,000 geochemical sampling location map (Sentinel-2 MSI red, green, and blue: bands 8, 4, and 2, respectively).

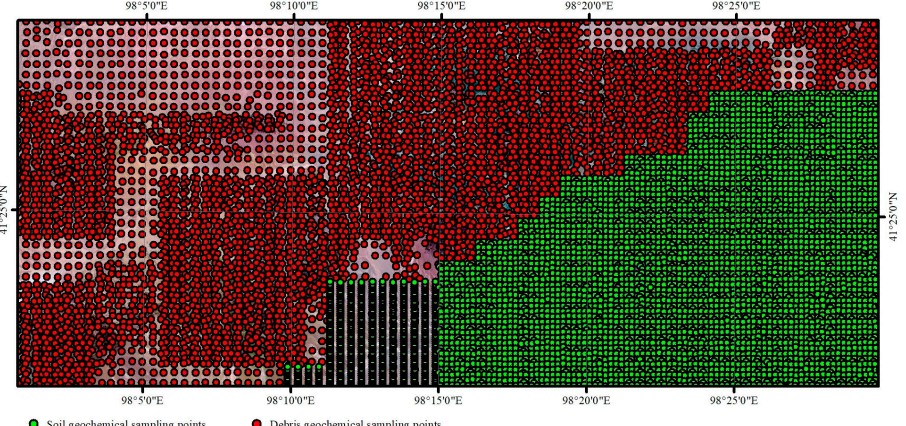

**Figure 5.** The 50,000 geochemical sampling location map (Sentinel-2 MSI red, green, and blue: bands 8, 4, and 2, respectively).

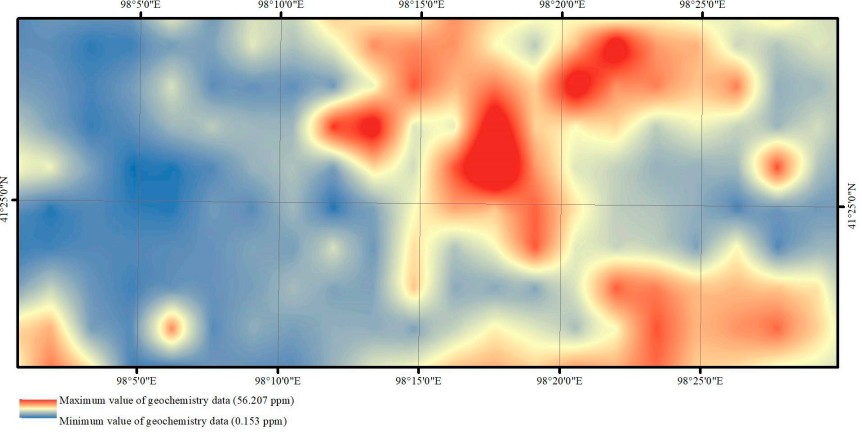

**Figure 6.** Map showing the spatial distribution of Cu modeled using the Kriging interpolation method (200,000).

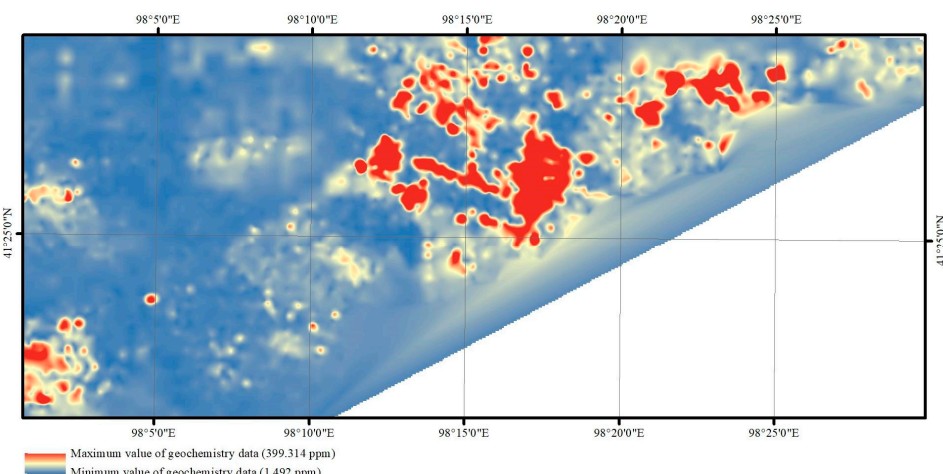

**Figure 7.** Map showing the spatial distribution of debris Cu modeled using the Kriging interpolation method (50,000).

*4.3. Regression Method*

As mentioned previously, remote sensing data were obtained by detecting the reflection of objects on the Earth's surface from solar radiation. These data can provide information on lithology, geological structure, and alteration through various wave bands, with different remote sensing sources offering complementary advantages and correlations between features. Remote sensing images offer a wide coverage area, intuitive visuals, and rich comprehensive data. These images allow for direct extraction of geological features, including strata, lithology, structure, and quaternary distribution, as well as bare rocks, topographic and geomorphic features, and water system distribution. However, geochemical data represent the spatial distribution of chemical elements, with geochemical indexes indicating the main chemical elements present on the Earth. Geochemical exploration methods rely on chemical elemental anomaly information as a crucial indicator for direct prospecting.

In this study, the method used for image fusion is pixel-level fusion. This method directly processes the image to obtain detailed information by fusing the original layers. Pixel-level fusion can be divided into two categories, including spatial and transform domain-based fusion methods. Spatial domain-based fusion directly uses pixel values for fusion, while the transform domain-based fusion transforms the image into coefficients and combines the obtained individual coefficients to achieve fusion. The fusion method used in this paper is based on a linear regression model, which incorporates the spectral and spatial characteristics of the remote sensing and geochemical data and is a spatial domain-based fusion method. The proposed method produces fused geochemical layers with higher spatial resolution, which can potentially provide a comprehensive understanding of the distribution of geochemical elements in geographic areas, making it a useful tool for geological and mineral exploration.

To improve the resolution of the geochemical elemental layer, the method used in this study involves several steps. Firstly, the geochemical elemental layer and corresponding areas of remote sensing images are obtained and assessed. Secondly, preprocessing of the remote sensing image is done, followed by clustering and alteration information acquisition. Thirdly, alteration information is acquired for image decomposition into low-frequency and high-frequency images. Fourthly, within the scope of ground objects in each cluster, the pending geochemical element layer and the first graphic first function relation are obtained. Finally, the first function relation and the secondary graphic spatial details and the pending geochemical elemental layer are used to obtain a high-resolution layer. The purpose is to achieve artificial sampling without any increase in the cost and difficulty of sampling, based on the improved resolution of the geochemical element layer.

According to the above characteristics, this paper selects logistic regression methods for fusion research, including global and local logistic regression methods. The overall technical process is shown in Figure 8.

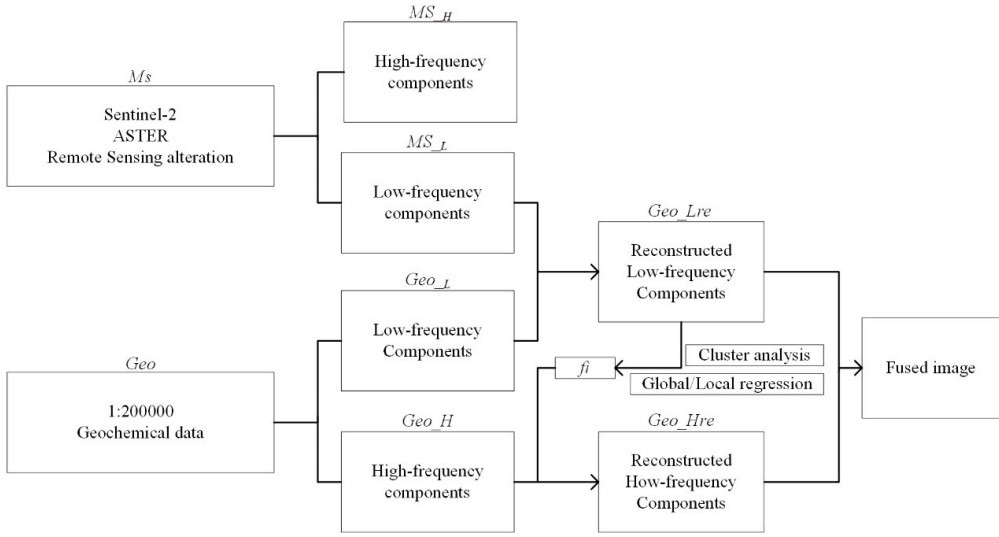

**Figure 8.** Methodological flowchart.

1.    Image decomposition

The remote sensing images and alteration information images (MS) are used for data fusion. The image Laplacian pyramid algorithm is used to decompose the remote sensing images and alteration information images into a low-frequency image (MS_L) and a high-frequency image (MS_H) with respect to the spatial details and background, respectively.

2.    Geochemistry data scaling transformation

As the geochemical elemental layers (Geo) are the result of interpolated data at interval sampling points, the geochemical element layers need to be resampled by cubic convolution to the same resolution as the remote sensing or alteration information images. The resampled geochemical data are similarly decomposed into two parts: a low-frequency image (Geo_L) and a high-frequency image (Geo_H).

3.    Cluster analysis

Cluster analysis is a process that divides a concrete or abstract data set into several groups or classes. This technique is widely used in data mining, image segmentation, pattern recognition, spatial remote sensing technology, feature extraction, signal compression, and many other fields and has obtained many satisfactory results [38,39]. Cluster analysis identifies clustering structures in a dataset that are characterized by the maximum similarity within the same cluster and the maximum dissimilarity between different clusters [40,41]. In this case, the similarity between similar datasets is as large as possible, and the similarity between dissimilar data sets is as small as possible. Cluster analysis is required for local regression fusion. In this paper, a K-means clustering algorithm is used for the unsupervised classification of all the geological bodies contained in remote sensing images and alteration information images. N classes of clustering centers, each of which represents a feature range of a geological body, where $N \geq 2$, are obtained.

The K-means algorithm is generally the most well-known and widely used clustering method. Various extensions of K-means have been proposed in the literature [38]. The K-means clustering algorithm is a hard clustering algorithm that is representative of the prototype-based clustering method of objective functions. The algorithm involves the distance between data points and uses the prototype as the optimized objective function. The optimization rules of iterative operation can be obtained by using the method of finding

the extremum of the function. The K-means clustering algorithm takes the Euclidean distance as a similarity measure to find the optimal classification of the center vector V of an initial cluster to minimize the evaluation index J. The algorithm uses the error sum of squares criterion function as the clustering criterion function [42,43].

The process of the K-means clustering algorithm is as follows. First, the initial clustering center is determined, i.e., K points are randomly selected from the dataset. Second, the distance from each sample point to the selected clustering center is calculated according to a distance formula, and according to the minimum distance criterion, each sample point is classified into the class represented by the clustering center with the shortest distance from the point. Then, the average value of the objects in each class is calculated and compared with the last calculation. If there is no change in the clustering center between these two calculations or the change does not exceed the given threshold, it means that the clustering criterion function converges and the clustering algorithm ends. Otherwise, we continue to adjust the clustering centers and restart the next iteration. In one iteration of the algorithm, the clustering center does not change, and the class to which each sample point belongs no longer changes, which means that all samples are correctly classified and the algorithm ends [44,45].

4. Establishment of the correlation function

Third, a multiple linear regression function is established independently for the global region or every cluster by considering the linear relationship between MS_L and Geo_L. According to the correlation between the concentrations of Geo_L and MS_L, reconstructed fused low-frequency geochemical data (Geo_Lre) are obtained.

The expression of the linear function relation is as follows:

$$Geo\_Lre = f_i(MS\_L_1, MS\_L_2, \ldots\ldots, MS\_L_n), \tag{1}$$

where i is the total number of bands in the MS image or alteration information image and n (n = 1, 2, . . . , i) is the band number. The correlation coefficients and constants of the function can be calculated using the least squares approach.

According to the linear function, the spatial details of MS_H are injected into Geo_H to be processed within the feature range of each geological body to obtain the target geochemical elemental layer. The correlation coefficients and constants of the linear function are used to obtain the reconstructed fused high-frequency geochemical data (Geo_Hre) as follows:

$$Geo\_Hre = Geo\_H + f_i(MS\_H_1, MS\_H_2, \ldots\ldots, MS\_H_n), \tag{2}$$

On the basis of the above example, in the application of an implemented case as shown in Figures 2 and 3, each category of different geological body units within the scope of the first function is described based on the relationship. The categories are described in the second image space based on the details mentioned above, including the geochemical elemental layer, to achieve the target geochemical elements. It is a linear model within different geological body units, and the detailed information of high frequency images is fully considered to further improve the accuracy of subsequent fusion.

5. Image fusion

The high-resolution fusion image Geo_f can be obtained by merging the resampled geochemical elemental layer Geo_Lre with the fused high-frequency image Geo_Lre, as follows:

$$Geo\_f = Geo\_Lre + Geo\_Hre, \tag{3}$$

### 4.4. Image Fusion Quality Evaluation

The evaluation of the quality of remote sensing image fusion results includes two main aspects: subjective evaluation and objective evaluation. The subjective evaluation method involves researchers directly observing the fused images with the naked eye and assessing the quality of the fusion based on their subjective experience. In this study, researchers can assess the quality of the fusion by examining whether mineralized points are found during

field surveys. The objective evaluation method involves using quantitative formulas such as information entropy, spectral information, and correlation coefficients to evaluate the fused image information. In this study, the evaluated 50,000 geochemical mapping data points serve as the basis for both the subjective and objective evaluation of the quality of image fusion.

## 5. Research and Discussion

Sentinel-2 data are used as the original images for fusion because of the high signal-to-noise ratio among the free remote sensing data. ASTER data are used as another remote sensing data source for fusion because of their rich shortwave infrared band settings that can provide remote sensing alteration information. The ASTER Mineral Index Processing Manual by Aleks Kalinowski and Simon Oliver (http://www.ga.gov.au/image_cache/GA7833.pdf, accessed on 26 February 2023) provides a comprehensive overview of the mineral indices derived from ASTER data. These authors suggest a range of band combinations and ratios for mapping various mineral assemblages in relation to different styles of alteration (Table 1).

**Table 1.** ASTER band ratios for enhancing mineral features.

| Mineral Feature | ASTER Band Combination(s) |
| --- | --- |
| Ferric iron | 2/1 |
| Ferrous iron | 5/3 and 1/2 |
| Ferric oxide | 4/3 |
| Gossan | 4/2 |
| Carbonate/chlorite/epidote | $(7 + 9)/8$ |
| Epidote/Chlorite/Amphibole | $(6 + 9)/(7 + 8)$ |
| Amphibole | $(6 + 9)/8$ and $6/8$ |
| Dolomite | $(6 + 8)/7$ |
| Carbonate | 13/14 |
| Sericite/Muscovite/Illite/Smectite | $(5 + 7)/6$ |
| Alunite/Kaolinite/Pyrophyllite | $(4 + 6)/5$ |
| Phengite | 5/6 |
| Kaolinite | 7/5 |
| Silica | 11/10, 11/12, 13/10 |
| SiO$_2$ | 13/12, 12/13 |
| Siliceous rocks | $(11 \times 11)/(10 \times 12)$ |

Several band ratios have also been proposed to map mineral indices (Table 2) [21,46].

**Table 2.** ASTER false color composites for enhancing mineral features.

| Mineral Feature | ASTER Band Combination(s) |
| --- | --- |
| Silica index | band 11/band 10, band 11/band 12, band 13/band 10 |
| Biotite-epidote–chlorite–amphibole index | (band 6 + band 9)/(band 7 + band 8) |
| Skarn carbonates–epidote index | (band 6 + band 9)/(band 7 + band 8), band 13/band 14 |
| Garnets-pyroxenes index | band 12/band 13 |
| Iron oxide index | band 2/band 1 |
| White micas Al-OH depth | (band 5 + band 7)/band 6 |
| Carbonates Mg-OH depth | (band 6 + band 9)/(band 7 + band 8) |
| Carbonate abundance | band13/band14 |

To better obtain the fusion relationship between alteration information and geochemistry, a principal component transformation was performed on all the extracted alteration information that was obtained from ASTER data. This was done because ASTER data provide diverse alteration information. PC2 was selected from the principal component analysis as it concentrates the information of the maximum relevant variability outside the

two factors and more reflects the alteration information. The same approach was applied to the Sentinel-2 data which has rich multispectral bands. In the principal component analysis, PC1 reflects the weighted information of all the bands, which includes the background values and topographic factors. Therefore, PC2 was chosen as the data source for fusion because it concentrates on the most relevant information regarding alteration information and geochemistry.

Figures 9–12 demonstrate the results of Sentinel-2 data PC2 global regression fusion, Sentinel-2 data PC2 local regression fusion, ASTER alteration information PC2 global regression fusion, and ASTER alteration information PC2 local regression fusion. Overall, global regression is better at providing the range of anomalies in small-scale geochemical data, while local regression cannot show relevant information. The global regression fusion results give a more precise prediction of a relatively concentrated and small number of new anomalies with a wider area of individual anomalies. Contrarily, the local regression fusion results provide a larger and more dispersed number of new anomalies with a smaller area of individual anomalies, giving more detailed information on new anomalies for mineral exploration prediction. Visually inspecting the results, both features have their advantages. The global regression fusion results display new anomalies within the original small-scale geochemical anomalies, whereas the local fusion results provide richer and finer information on new anomalies.

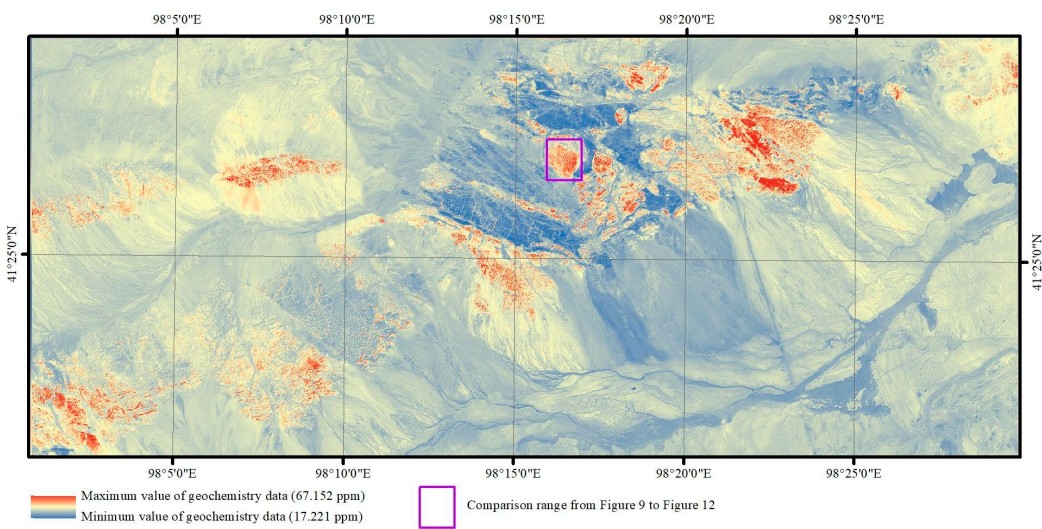

Maximum value of geochemistry data (67.152 ppm)
Minimum value of geochemistry data (17.221 ppm)

Comparison range from Figure 9 to Figure 12

**Figure 9.** Fusion image of Sentinel-2 data PC2 local regression.

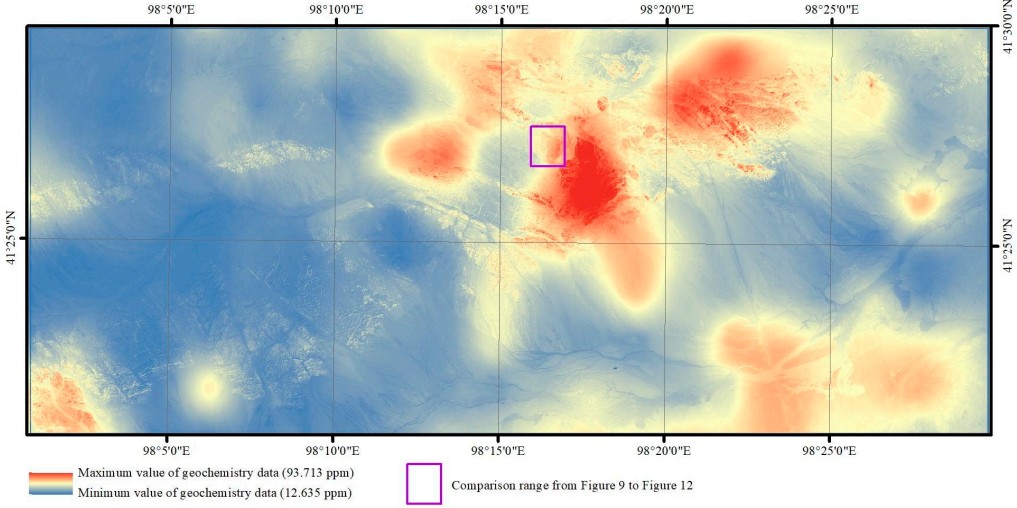

Maximum value of geochemistry data (93.713 ppm)
Minimum value of geochemistry data (12.635 ppm)

Comparison range from Figure 9 to Figure 12

**Figure 10.** Fusion image of Sentinel-2 data PC2 global regression.

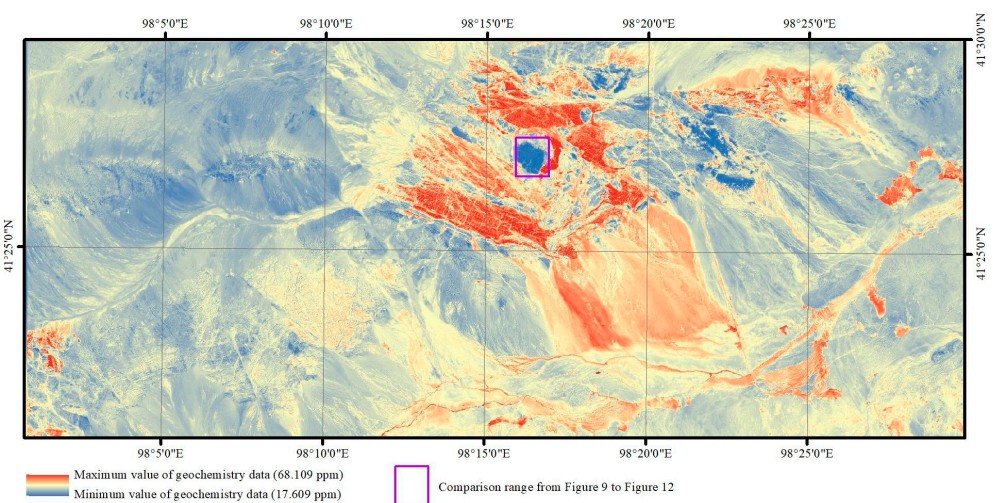

Maximum value of geochemistry data (68.109 ppm)
Minimum value of geochemistry data (17.609 ppm)

Comparison range from Figure 9 to Figure 12

**Figure 11.** Fusion image of ASTER alteration information PC2 local regression.

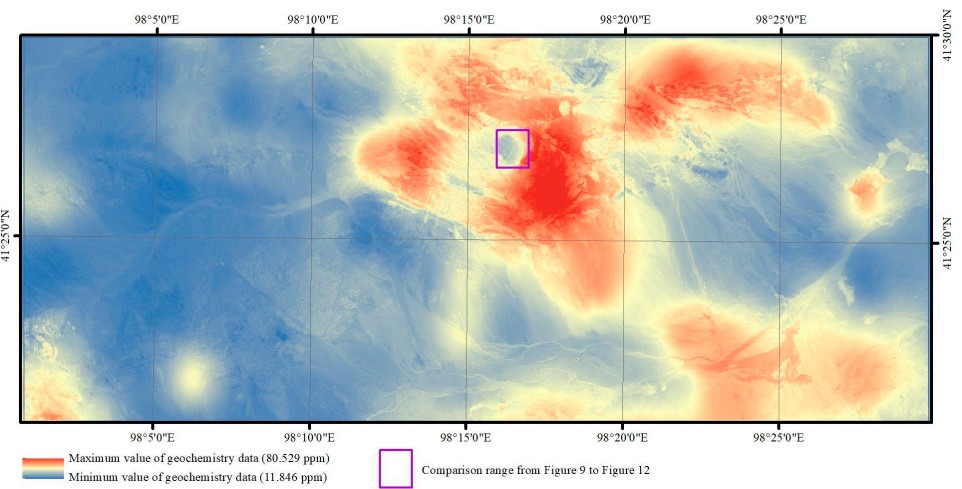

Maximum value of geochemistry data (80.529 ppm)
Minimum value of geochemistry data (11.846 ppm)

Comparison range from Figure 9 to Figure 12

**Figure 12.** Fusion image of ASTER alteration information PC2 global regression.

A comparison of the fusion results from the original image and the alteration information, the fusion results of band PC2 of the original image and band PC2 of the alteration information reveals a certain negative correlation between the two fusion results, such as the areas shown in the purple boxes in Figures 9–12. Compared with the 1:50,000-scale geochemical data, the fusion results of the original image band PC2 show a somewhat negative correlation, while the fusion results of alteration information PC2 show a somewhat positive correlation. The results of the original image fusion are shown as medium-high values in yellow and red colors, while the results of the alternation fusion are shown as low values in blue color.

Magnifying the distribution of the largest alteration anomalies in the middle position of the 1:200,000-scale geochemical data layer can more clearly yield the above-described conclusions (Figure 13a–c). Local regression fusion can provide a range of anomalies in small-scale geochemical data, and it may be more effective to screen new anomalies in the range of 1:200,000-scale geochemical anomalies. However, we also see that the 1:200,000-scale and 1:50,000-scale geochemical anomalies are not completely consistent, and possible new anomalies that are found only in the range of 1:200,000-scale geochemical anomalies may also lead to the omission of valid information. The local images also show that the results that were obtained from the PC2 fusion of alteration information are more scattered, with more anomalies and smaller areas. In contrast, the fusion results obtained from the original image PC2 are more concentrated and have fewer anomalies and larger areas. The positive and negative correlations are also consistent with the previous analysis;

i.e., the fusion results of the original image band PC2 show a somewhat negative correlation (Figure 13c,d), while the fusion results of the PC2 etched information PC2 show a somewhat positive correlation (Figure 13e,f).

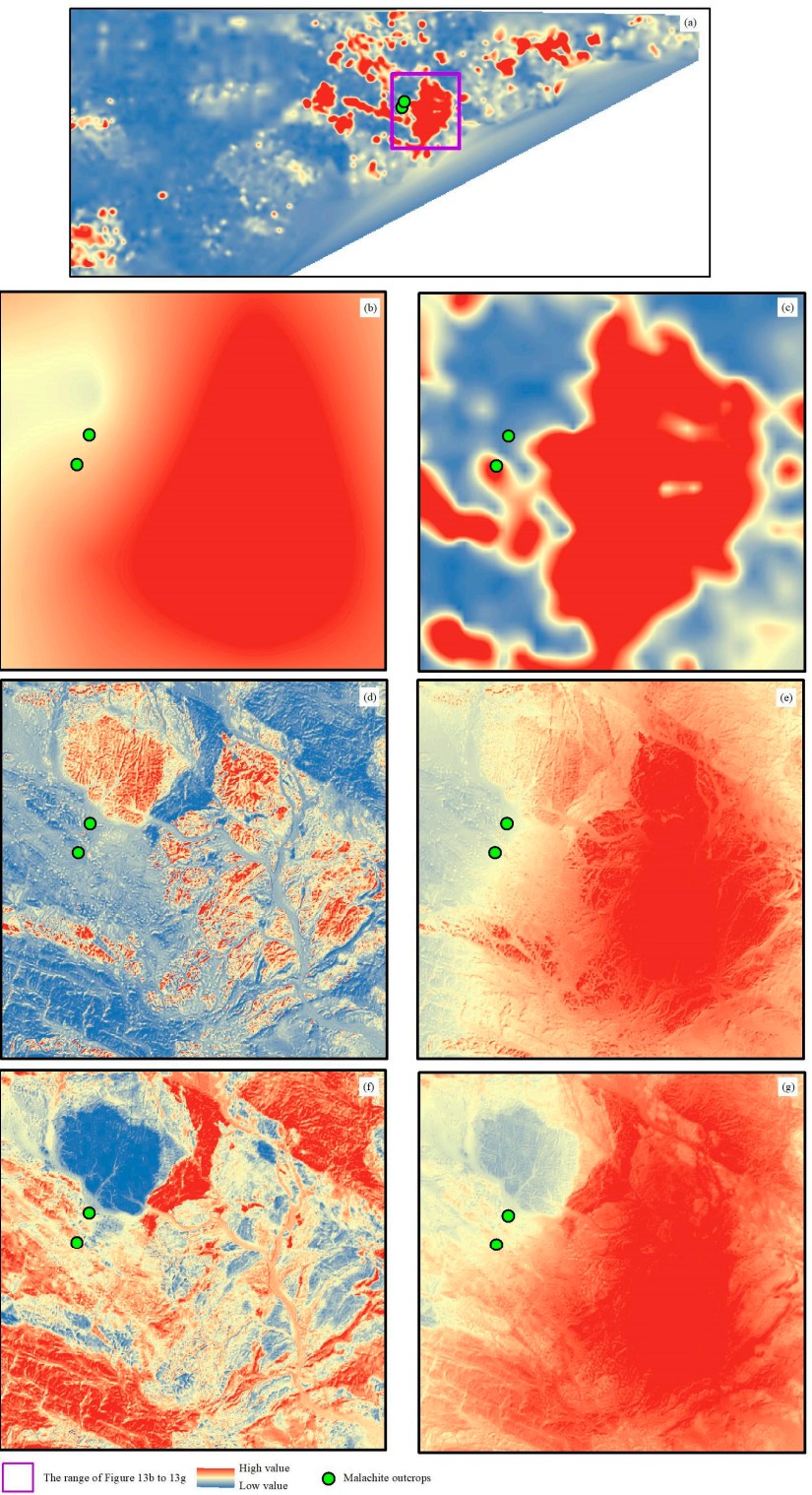

**Figure 13.** Local comparison image of: (**a**) the comparison range of (**a–g**); (**b**) 200,000-scale geochemistry; (**c**) 50,000-scale geochemistry; (**d**) Sentinel-2 data PC2 local regression fusion; (**e**) Sentinel-2 data PC2 global regression fusion; (**f**) ASTER alteration information PC2 local regression fusion; and (**g**) ASTER alteration information PC2 global regression fusion.

To verify the quantitative analysis, see Table 3, which contains the correlation between the original image fusion results, the fusion results of the alteration information, and the 1:200,000-scale geochemical data. Due to the complexity of geochemical data filling, the correlation between the 1:50,000- and 1:200,000-scale geochemical data also did not show an obvious positive correlation, where the correlation coefficient is −0.3. The highest correlation among the fusion results is the regression fusion result of local alteration information, which is higher than the correlation between the 1:50,000- and 1:200,000-scale geochemical data. The correlation results also show that original image band PC2 shows a positive correlation with the 1:200,000-scale geochemical data, while the PC2 alteration information shows a negative correlation with the 1:200,000-scale geochemical data. The negative correlation of the fusion results of the alteration information may be related to the fact that too many types of alteration information were extracted, the geological background corresponding to each type of alteration information was not analyzed, and the principal component analysis was performed directly. The subsequent detailed analysis and fusion of each type of alteration information is needed to find the optimal combination. In fact, a negative correlation is also shown between the 1:200,000 and 1:50,000 data. This indicates that geochemical anomalies are a very complex correspondence that is influenced by a variety of factors such as sampling site location, density, and sampling method.

**Table 3.** Correlation coefficient of different fusion methods.

|  | Sentinel-2 Bands PC2 | ASTER Alteration Information PC2 |
| --- | --- | --- |
| Global | 0.12 | −0.17 |
| Local | 0.27 | −0.32 |

In the field survey, which is the most effective means to verify the indoor interpretation results, we found several malachite outcrops (Figure 14). In the 1:200,000 geochemical layer, malachite outcrops are located at the edge of the geochemical high value (Figure 13a), while in the 1:50,000 geochemical layer, one malachite outcrop is located in the geochemical high value area and another malachite outcrop is located in the geochemical low value area (Figure 13b). In the fused image, malachite outcrops are in the high value area in the ASTER alteration information fusion results (Figure 13c,d), while they are in the low value area or the edge of high value area in the original image fusion results (Figure 13e,f). From the field survey results, the fused results, especially the large-scale geochemical layers obtained by fusing the alteration information, can provide some high value information that cannot be shown in the small-scale geochemical layers.

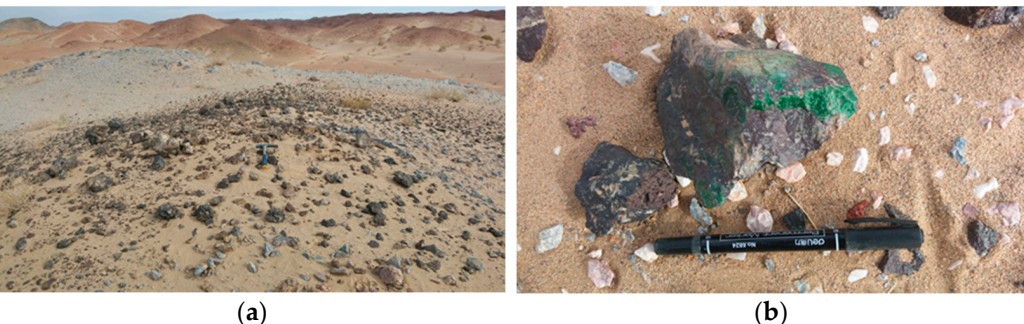

(a)　　　　　　　　　　　　　　　　(b)

**Figure 14.** Photographs of malachite outcrops and hand specimens: (**a**) malachite outcrops; (**b**) malachite hand specimens.

## 6. Conclusions

In this paper, we establish a linear regression equation for data fusion between remote sensing images and geochemical data. Remote sensing raw images and remote sensing alteration information are fused with small-scale geochemistry to obtain fused large-scale geochemical inversion data, which provide a source of data analysis for mineral exploration

in areas lacking large-scale geochemical data. The regression equation is established using the low-frequency images obtained from image decomposition; then, the detailed spatial information on the high-frequency images is injected for fusion and the obtained fusion results achieve better validation results in quantitative correlation coefficients, visual observation, and field sampling. The fusion results of Sentinel-2 raw images show positive correlations, while the fusion results of ASTER alteration information show negative correlations. Global regression fusion can provide a range of anomalies in small-scale geochemical data, and it may be more effective to screen new anomalies in the range of 200,000 geochemical anomalies. This equation provides a new idea for data source acquisition for regional geological and mineral exploration in areas lacking large-scale geochemical data. More fusion models should be used in the future to explore the most effective fusion method. This method will be especially beneficial to mineral exploration work in high-altitude uninhabited areas, such as Xinjiang and Tibet in western China.

**Author Contributions: Conceptualization:** H.D. and L.J.; methodology, H.D. and M.X.; software, S.B.; validation, C.Y., L.L. and H.D.; writing—original draft preparation, H.D. and M.X.; writing—review and editing, L.J.; visualization, C.Y. and L.L. All authors have read and agreed to the published version of the manuscript.

**Funding:** This study was supported by the Second Tibetan Plateau Scientific Expedition and Research (STEP), (2019QZKK0806) and the National Natural Science Fund of China (41972308).

**Data Availability Statement:** Not applicable.

**Conflicts of Interest:** The authors declare no conflict of interest.

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
