# Peer review of "Research on Scale Improvement of Geochemical Exploration Based on Remote Sensing Image Fusion"

_remotesensing, doi:10.3390/rs15081993_

Round 1

Reviewer 1 Report

Review of the manuscript

Research on Scale Improvement of Geochemical Exploration 2 Based on Remote Sensing Image Fusion

Manuscript ID: remotesensing-2278737

The current work integrated the fused remote sensing images with small-scale geochemical data based on a linear regression model, aiming to improve the resolution of geochemical elemental layers and provide reference data for mineral exploration in areas lacking large-scale geochemical data. The manuscript in the present for is not recommended to publish in Remote Sensing, requiring major revisions. However, several issues should be considered. My main comments are listed below:

The manuscript has the following major issues:

1. It is not written clearly using correct English grammar and syntax, containing many unclear and confusing sentences. I strongly suggest that the manuscript should be revised to improve its English.

2. a considerable jump in the text and you suddenly speak about issue without any background such as the PC2 (page 11). You mention it without any information to the readers what is it!!

3. The manuscript is not organized well, some figures inserted in the file before the authors refer to them in the text. There are figures added without any clear significance such as Figures 4 and 5.

4. The manuscript needs to be thoroughly edited for accuracy, clarity and consistency.

5. All Tables in paper with no significance to the title and the authors did not use it in there analysis.

6. The conclusion need to be realistic and supported by your new data.

7. Figures need to be improved and clarified. Some of them should be deleted.

Detail comments:

·         Page 3, Line 121: Figure 1, please review the figure in the revised PDF file.

·         Page 4, Lines 140, 143: Figures 2 and 3, the authors did not refer to them in the text. And I have a question the purple box in all figures refer to you area of interest, why authors did not male clip or subset with this box to make it the borders of the area?

·         Page 5 Line 176: Table 1 should be deleted there is no significance.

·         Page 6 Lines 187, 195: Figures 3 and 4 what do you want to say with these figures?? what is the value or information?? you do not refer for them in the text and there are not any comments help the topic added.

·         Page 6, Line 206: … distribution of Cu with Kriging … what do you mean with Cu.. are you mean concentration or what please explain.

·         Page 7, Line 211: Figure 6, you should write the values of Cu low and high geochemistry data on the legend. In the present case it is general speaking.

·         Pages 10 and 11, Lines 344, 346, Tables 2 and 3, the authors did not use any of ratios mentioned in these table, although what the benefit of these to tables in the manuscript.

·         Page 11, line 350, you mentioned the PCs and after that Pc1 without any introductory about the principal component analysis and how you can get these PCs. This can make a considerable confusion to the readers.

·         Page 12, 13: Figures 9-12 should specify the values of low and high in the legend.

·         Page 13, Line 382: Figure 13, I suggest to add index map to make it easy to the readers to know the geographic locations of these subsets.

·         Page 13, Line 391: the authors mentioned The positive and negative correlations … and then linear regression ….. my question where is the figure or relation represent this linear regression? Where are the X-Y figure shows this relation??

·         There are another detailed comments in the PDF file, Please response to them

Finally, the paper may be publishable in remote sensing  after major revisions. The authors are encouraging to response with the above comments/others in the attached PDF version.

Thank you for your patience

Reviewer 2 Report

Reviewer Blind Comments to Author:

This research paper (remotesensing-2278737-peer-review-v1) brings improvement of mineral exploration techniques with integration of geochemical and remote sensing data. The proposed methods are based on geochemical dataset of rocks and remote sensing images of the VIS, VNIR, SWIR and TIR, which provide a robust and reliable distribution pattern of metal content in the area of study. The paper is one of the significant contributions to understanding the image decomposition of remote sensing data into low-frequency and high frequency band with respect to the geochemical data.

The substantial development in remote sensing and geochemical exploration using decomposition techniques improve the resolution of geochemical mapping in mineral exploration. Thus, it is one of the most suitable for publication in Remote Sensing. However, this article brought criticisms that needs to additional reconsideration.

Abstract: There are significant corrections needed concerning the technical writing of the Abstract. Some corrections have been made to the pdf file.

Some issues need to be clarified as follows:

1.   Introduce the significance of geochemical data in mineral exploration.

2.   Give the basic information about the geological setting with field photographs used in your study area.

3.   Frequent generalized statements are used in this article, which needs to be removed.

4.   Finally, all points should be considering before proposal of your revised version of the paper to be submitted.

I noticed; this script should be suitable to recommend for publication in the Remote Sensing with minor modification.

Round 2

Reviewer 1 Report

Dear Authors

The manuscript remotesensing-2278737 titled " Research on Scale Improvement of Geochemical Exploration 2 Based on Remote Sensing Image Fusion"  has been significantly improved.

Authors have complied with the suggestions and corrections provided. Research highlights are much better discussed in the text and outlined as instructed. Figures are now much more clearly described and some ambiguities were clarified. The English language is also at a quite good level, especially after the authors made some necessary corrections. I thank the authors for their efforts addressing the comments raised in the reviews.

I recommend the manuscript be published in its present form in Remote Sensing after the Editor's satisfaction.

Thank you